A novel framework for secure cryptocurrency transactions using quantum crypto guard

http://orcid.org/0000-0002-9768-4925 Alsayaydeh Jamil Abedalrahim Jamil 1 jamil@utem.edu.my
http://orcid.org/0000-0001-8464-3932 Yusof Mohd Faizal 2 3
Yahaya Nor Adnan 3
Kovtun Viacheslav 4
http://orcid.org/0000-0002-5725-8075 Herawan Safarudin Gazali 5
1 Department of Engineering Technology, Fakulti Teknologi & Kejuruteraan Elektronik & Komputer (FTKEK), Universiti Teknikal Malaysia Melaka (UTeM) , Melaka , Malaysia
2 Research Section, Faculty of Resilience, Rabdan Academy , Abu Dhabi , United Arab Emirates
3 Centre of Postgraduate Studies, University Malaysia of Computer Science and Engineering , Selangor , Malaysia
4 Department of Computer Control Systems, Vinnytsia National Technical University , Vinnytsia , Ukraine
5 Industrial Engineering Department, Faculty of Engineering, Bina Nusantara University , Jakarta , Indonesia
Bhattacharyya Siddhartha
Electronic publication date: 2025 Sep 12
Publication date: 2025
Volume: 11
Electronic Location ID: e3030
Received 2024 Nov 14; Accepted 2025 Jun 23
Copyright: © 2025 Alsayaydeh et al.
Copyright year: 2025
Copyright holder: Alsayaydeh et al.
License: This is an open access article distributed under the terms of the Creative Commons Attribution License, which permits unrestricted use, distribution, reproduction and adaptation in any medium and for any purpose provided that it is properly attributed. For attribution, the original author(s), title, publication source (PeerJ Computer Science) and either DOI or URL of the article must be cited.
License URL: https://creativecommons.org/licenses/by/4.0/

Keywords: Cryptocurrencies, Cryptographic systems, Blockchain technology, Security, Zero knowledge proof

Funding: The authors received no funding for this work.

==============================
In today’s digital world, cryptocurrencies like Bitcoin can secure transactions without banks. However, the rise of quantum computing poses significant threats to their security, as traditional cryptographic methods may be easily compromised. In addition, the existing algorithms face difficulties like slow transaction speeds, interoperability issues between different cryptocurrencies, and privacy concerns. Hence, Quantum Crypto Guard for Secure Transactions (QCG-ST), a novel blockchain framework, is introduced, offering enhanced security and efficiency for cryptocurrency transactions. The QCG-ST employs lattice-based cryptography to provide robust protection against quantum threats and incorporates a new consensus mechanism to increase the transaction speed and reduce energy consumption. The QCG-ST system uses lattice-based encryption that is based on the Ring Learning With Errors (Ring-LWE) issue to protect itself from quantum assaults. It uses sharding, a Proof-of-Stake (PoS) consensus method, and a threshold signature scheme (TSS) to make the system more scalable and use less energy. Zero-knowledge proofs (ZKPs) are used to check transactions without giving out private information. We offer a cross-chain atomic swap protocol that uses hashed time-lock contracts to make sure that it works on all platforms. Blockchain transaction data utilized in testing originated from the Bitcoin Historical Dataset available on Kaggle, and quantum resistance has been assessed using the Qiskit Aer simulator. It evaluated the framework’s performance to that of traditional methods like Payment Channel–Lightning Network (PC-LN), Variational Quantum Eigensolver (VQE), and Cross-Chain Transaction with Hyperledger (CCT-H). Results show that QCG-ST does far better than traditional systems in terms of transaction success rate (up to 98.5%), speed, energy efficiency, latency, and throughput, especially when tested in a quantum-simulated environment. This study completes in an essential vacuum in blockchain technology by suggesting a strong, quantum-resistant, privacy-protecting architecture that can handle the problems that could arise up in decentralized digital banking in the future.

Introduction

The emergence of cryptocurrencies has brought about a revolution in the digital era, enabling decentralized manner, and secure exchanges without the need for any intermediaries (Rushita, Sood & Yadav, 2023). The growth of cryptocurrencies has led to a wide range of digital currencies and blockchain applications. With the rising computing capability of quantum computing, security mechanisms placed cryptocurrency transactions at risk (Shah et al., 2023). The existing cryptographic techniques, are possesed to attacks and this vulnerability could able to compromise cryptocurrency transactions’ privacy, and security (Tom et al., 2023). The security of blockchain cryptosystems currently in use is susceptible to a higher level of security risk due to the recent quantum computing improvement (Dey, Ghosh & Chakrabarti, 2022). Preventing this attack in cryptocurrency transactions with blockchains platform results in the emerge of post-quantum cryptography (Gharavi, Granjal & Monteiro, 2024) models. The utilization of mathematical problems that are challenging for quantum computing systems, such as multivariate signature algorithms and lattice-based challenges, are faced to be secure against both conventional and quantum computers (Aydeger et al., 2024). These lattices based crypto challenges with learning error analysis highlights the demands for creating cryptographic solutions that works in the quantum technology (Sahu & Mazumdar, 2024). Quantum computers are an enormous risk to the traditional cryptography techniques that most cryptocurrencies use. For example, Shor’s technique can quickly factor enormous amounts and find discrete logarithms. These are the building blocks of Rivest–Shamir–Adleman (RSA) and elliptic curve cryptography (ECC), which are commonly employed in blockchain technology to protect wallets and confirm transactions. If a quantum computer became strong enough to perform Shor’s algorithm, it could destroy these systems. This would let an attacker get private keys from public keys and put the whole bitcoin infrastructure at risk. Also, Grover’s technique speeds up brute-force attacks on symmetric cryptographic systems by a factor of two, which means that algorithms like SHA-256 used in Bitcoin’s Proof-of-Work have only half as secure at the bit level. To make sure that blockchain systems are safe and strong against new quantum threats, we need to use post-quantum cryptography methods like lattice-based schemes.

Purpose and scope

Beyond security concerns, conventional blockchain technologies face several challenges. These include slow transaction speeds, high energy consumption due to resource-intensive consensus mechanisms like Proof-of-Work, interoperability issues between various cryptocurrencies, and maintaining privacy within decentralized systems (Habib et al., 2022). Addressing these issues has led to growing research interest in quantum-resistant and scalable blockchain solutions (Dzidzikashvili & Kheladze, 2022). However, the efforts to promote cross-chain interoperability solutions and achieve compatibility across diverse systems must be focused on Belchior et al. (2021). Minimizing the chances of hacking and privacy breaches, atomic cross-chain swaps allow numerous parties to exchange assets across various blockchains directly without intermediaries (Mohanty et al., 2022). The evaluation of transaction records captures the anonymity, though the research uses linkable ring-based signature verification with appropriate key size and time costs (Zhang et al., 2024).

Existing signature systems don’t simultaneously cover regular and contract transactions, leaving them vulnerable to malleability attacks (Feng et al., 2023). The consensus process validates transactions and adds blocks while assuring credibility, consistency, decentralized management, and openness, whereas Blockchain goals determine consensus mechanism selection like Proof-of-Stake (PoS) (Yusoff, Hasan & Abd Ali, 2024). These consensus processes, like sharding, layered solutions, and PoS, aim to enhance blockchain scalability and security compared to existing Proof of Work (PoW) while minimizing energy consumption (Junaidi et al., 2023). In quantum computing, quantum-assisted cryptocurrency transactions are analyzed using quantum bits as a state vector called superposition scale (Sinai & Peter, 2024; Joshi, Choudhury & Minu, 2023).

Research novelty

The study presents the Quantum Crypto Guard for Secure Transactions (QCG-ST) architecture, which combines quantum-resistant cryptographic algorithms with data from blockchain networks to make bitcoin transactions safer. At first, the study uses lattice-based cryptography and Ring-based Learning With Errors (Ring-LWE) to protect against known quantum attacks. The framework uses a new way of reaching agreement called Proof-of-Stake (PoS), which is meant to speed up transactions and lower the amount of energy that blockchain networks use overall. Zero knowledge proofs (ZKPs) are used for different types of transactions and to keep a cross-chain atomic swap protocol for multiple blockchains utilizing hashed time lock limitations. The blockchain framework makes sure that transactions are safe and that communication is clear between multiple nodes, just like secure home security systems that use GSM technology to find threats early (Alsayaydeh et al., 2021).

Contribution terms

The key focus of the research is as follows: Quantum Crypto Guard for Secure Transactions (QCG-ST)—to make bitcoin transactions secure and more rapidly, especially in light of new threats from quantum computing.

It employs the Ring Learning With Errors (Ring-LWE) approach to secure transaction data from quantum assaults by leveraging lattice-based encryption. This makes sure that the key generation and encoding operations are strong enough to be safe even against quantum attackers.

To add sharding and threshold signature schemes (TSS) to a Proof-of-Stake (PoS) protocol, which together make transactions faster, more scalable, and more energy-efficient across distributed nodes.

It creates a cross-chain atomic swap protocol that uses hashed time-lock contracts to make it easier for assets to move across different blockchain networks. This system lets chains engage with one other efficiently and securely without any intermediaries.

We use ZKPs to keep private information like transaction values and identities secure during the transaction verification process.

Document structure

The structure of the research article is arranged as follows: ‘Materials and Methods’ provides a literature review related to cryptocurrency security mechanisms with blockchain networks and implemented simulation environments. ‘Discussion’ implements the novel QCG-ST framework for secure asset transactions in a quantum computing setup. ‘Conclusions’ details the Qiskit_aer simulation parameters and performance analysis with metrics. Conclusion and future work concludes the research work.

Materials and Methods

Literature review

Grey & Chatib (2024) suggested using multi-signature wallets, improved consensus methods, and cryptographic approaches to make the system more trustworthy and secure. The impact of recent blockchain protocol innovations enhanced cryptocurrency security. The research shows substantial security gains using secondary data from security inspection and case reports. Still, it may not be able to inform much regarding how well or how scalable these innovations will be in the long run because it relies on already available information. Formal verification techniques have been applied to various domains to ensure system reliability. For instance, in power systems, model checking techniques have been used to verify system integrity under different conditions (Shkarupylo et al., 2022).

De Silva, Thakur & Breslin (2024) applied a payment channel (PC) based cryptocurrency card payment protocol that makes use of the Lightning Network (LN) and smart cards to bring down transaction fees and increase transaction speeds. Information about privacy comparisons, transaction throughput, and game-theoretical models of agent interactions are utilized in the study. Fewer operational expenses, faster settlement, and enhanced privacy are some benefits; technical complexity, possible difficulties in broad adoption, and government control are some drawbacks.

Starting with PoW, Jain, Gupta & Gupta (2025) some blockchain consensus algorithms, including Weighted Byzantine Fault Tolerance (WBFT), Proof of Elapsed Workload and Latency (PoEWAL), Reliable, Replicated, and Fault-Tolerant consensus algorithm (Raft), and Practical Byzantine Fault Tolerance (PBFT). The performance metrics, like throughput, latency, fault tolerance, and energy consumption, are compared with the protocols, drawing attention to their strengths and weaknesses. Testing on simulators like Hyperledger Fabric to fix PoW’s flaws and make it more secure, scalable, and performant. While WBFT stands out for thwarting corrupt nodes from joining the consensus, PBFT is known for its excellent throughput but poor scalability.

Carrascal et al. (2024) improved cryptocurrency arbitrage using the differential evolution (DE) optimization technique for the Variational Quantum Eigensolver (VQE) in the Qiskit framework. Convergence to optimal solutions is achieved in 417 steps on IBM’s real quantum processors with up to 127 qubits, proving that the DE-based technique is more reliable than classical optimizers like Constrained Optimization BY Linear Approximations (COBYLA). Although the findings suggest a possible quantum benefit in solving complicated arbitrage situations, drawbacks include the approach’s inability to scale to more extensive and complex problems and its reliance on particular quantum hardware.

Yadav (2023) employed lattice-based cryptography to resist quantum attacks that can solve lattice problems in Ring-based LWE schemes. Due to its hardness, these problems remain hard for classical and quantum computers. The study demonstrated that lattice-based cryptography provides secure transactions of cryptocurrencies, improves privacy and is suitable for post-quantum cryptography due to its robust security properties. Quantum computing power increases with the potential in post-quantum cryptography to avoid the classical cryptosystems breakage in polynomial time. Hence, Şahin & Akleylek (2023) proposed lattice-based cryptography with its potential worst-case hardness for group signatures schemes over lattices with fully dynamic schemes applied in blockchain technology.

Fallahpour (2024) discussed the cryptographic schemes in the post-quantum era, focused on the significant components of lattice-based cryptography called LWE problems and Fiat-Shamir transformation. Thus, the research analyzed these methods’ quantum resistance, affirming that lattice-based cryptography is considered quantum-safe due to its complexity for quantum adversaries. The study demonstrated that applied lattice-based cryptography is quantum-safe, but constraints on LWE instance sampling may still be vulnerable and require further analysis.

Rukhiran, Boonsong & Netinant (2024) examined the connection between GPU settings and system efficiency to improve GPU configurations for PoW blockchain mining, decreasing energy usage and improving transaction speed and finding substantial energy savings without sacrificing performance, using multilinear regression (MLR) to examine variables such as core speed, temperature, and fan speed. Experiments are conducted with different GPU models and mining software. The exclusive emphasis on particular software and hardware setups is one limitation that might restrict its generalizability.

Kumar, Selvarani & Vivekanandan (2023) suggested a cross-chain transaction (CCT) framework that uses quantum cryptography based on Hyperledger-based blockchains with signatures to improve transactions in terms of security, scalability, and interoperability. The simulation evaluates CPU utilization, communication overhead, block weight, and ledger memory, demonstrating how load diversification improves transaction processing rates. Limitations include the difficulty in integrating quantum cryptography and possible problems with scalability and resource optimization in the real world.

Blockchain interoperability and atomic cross-chain transactions are examined by Kotey et al. (2023) with an emphasis on essential techniques such as hashed time-lock contracts enabling atomic swaps. Research obstacles in implementing atomic swaps for frictionless cryptocurrency exchange are highlighted in this study, which also examines other blockchain interoperability initiatives.

The proposed QCG-ST architecture uses Ring-LWE because it has a solid base in the cryptographic research community and is relevant to ongoing attempts to set global standards. The National Institute of Standards and Technology (NIST) has been in charge of a multi-year initiative to create Post-Quantum Cryptography (PQC) standards. The goal of this project is to make cryptographic algorithms that can be used by the public that are resistant to quantum attacks. NIST has chosen Kyber, which is based on Module-LWE (a extension of Ring-LWE), as the key encapsulation mechanism (KEM) for standardization from among the finalists and selected schemes. This proves that the security assumptions of Ring-LWE and its variations are accurate. Other lattice-based systems show substantial quantum resistance, but we chose Ring-LWE because it corresponds with the cryptographic structure of Kyber and makes advantage of its known hardness assumptions and ease of implementation. Ring-LWE additionally has an algebraic structure that allows for small key sizes and fast computation, which makes it perfect for blockchain applications where speed and scalability are very important. So, choosing Ring-LWE is both an effective strategic move and supported by current attempts to harmonize it. This makes certain that the QCG-ST architecture follows quantum-safe principles that are recognized by the industry.

The existing research challenges related to cryptocurrency security, quantum computing risks, and blockchain interoperability are reviewed in this literature study. While secondary data restricts insights into scalability, some researchers focused on enhanced consensus procedures and multi-signature wallets. Though they acknowledge the difficulties in adoption, other works highlight how a payment channel protocol improves transaction speed and privacy. Others assess several consensus techniques and tackle scalability problems using differential evolution optimization applied to quantum computers for arbitrage. Despite their results’ potential lack of generalizability, some studies improve GPU mining settings. Despite presenting a quantum cryptography-based Cross-Chain Transaction with Hyperledger (CCT-H), their impacts run into problems with integration. Atomic cross-chain transactions are investigated to highlight the benefits and elaborate consensus problems. These significant difficulties need a novel solution for these research challenges, which can be resolved by implementing a novel QCG-ST framework. Table 1 shows the summary of reviewed literature based on qualitative factors.

Table 1 Summary of reviewed literature based on qualitative factors.

Ref.	Author(s)	Focus area	Approach/Protocol	Strengths	Limitations	Quantum resistance	
Grey & Chatib (2024)	Grey & Chatib	Conventional blockchain security	Multi-signature wallets, PoS	Enhanced trust and security	Limited scalability insights, uses secondary data	No	
De Silva, Thakur & Breslin (2024)	De Silva et al.	Transaction speed & Privacy	Payment Channel (PC), LN	Faster settlements, reduced fees, better privacy	High complexity, low adoption, regulation challenges	No	
Jain, Gupta & Gupta (2025)	Jain et al.	Consensus algorithm performance	WBFT, PBFT, PoEWAL, Raft	Evaluates multiple protocols on metrics	PBFT has poor scalability, dependent on simulator results	Partially (WBFT)	
Carrascal et al. (2024)	Carrascal et al.	Quantum optimization	VQE + DE in Qiskit	Fast convergence, real quantum hardware tested	Poor scalability, hardware dependent	Yes	
Yadav (2023)	Yadav et al.	Post-quantum security	Lattice-based (Ring-LWE)	High quantum resistance, improved privacy	Requires high computational resources	Yes	
Şahin & Akleylek (2023)	Şahin & Akleylek	Group signature security	Lattice-based schemes	Dynamic, post-quantum secure group signature schemes	Not yet widely deployed	Yes	
Fallahpour (2024)	Fallahpour	Cryptographic framework review	LWE, Fiat-Shamir Transform	Strong security proofs for quantum scenarios	LWE sampling constraints, theoretical bias	Yes	
Rukhiran, Boonsong & Netinant (2024)	Rukhiran et al.	Energy optimization in PoW	MLR on GPU configurations	Energy savings without performance loss	Limited generalizability to other platforms	No	
Kumar, Selvarani & Vivekanandan (2023)	Kumar et al.	Cross-chain framework	Hyperledger, Quantum Signatures	Scalable, interoperable, secure simulations	Complex integration, scalability concerns	Yes	
Kotey et al. (2023)	Kotey et al.	Atomic swaps & InteroperAbility	Hashed Time-Lock Contracts	Decentralized, low-cost token exchange	Multi-chain consensus challenges	No	

Research framework design

Data collection from blockchain networks

Initially, cryptocurrency transaction data must be collected from the blockchain network as block data by exchange and chain as blockchain nodes. The dataset used in this study, the Bitcoin Blockchain Historical Data has been obtained from open source Kaggle platform. The dataset can be accessed at https://www.kaggle.com/datasets/bigquery/bitcoin-blockchain?select=inputs (Kaggle). The data attribute includes information about sender and receiver account details, transaction amount on average, transaction time, and unique key used for the transaction. The attribute schema includes blocks and transactions for analysis. The blocks include (hash, size, stripped_size, weight) parameters for tracking block identity, size, and weight. The parameters (number, timestamp, version, merkle_root) provide block metadata, timestamp, and protocol details. The (nonce, bits) parameters define the mining difficulty and consensus process. The transaction_count represents the total number of transactions in the block. The transactions field includes (hash, size, virtual_size, and fee) and identifies transactions, size, and fees. The flow of cryptocurrency within the blockchain is given as sources (inputs) and destinations (outputs) of funds, with transaction value representing the flow of money. Also, the parameter is_coinbase flags the generation of new cryptocurrency as mining rewards. The time_lock, block_number/timestamp provides timing details and confirmation status about the secure transactions of cryptocurrencies.

The authenticity of the dataset is improved by filtering the witness transaction with a count between 500 and 3,000. Too few transactions under 500 may not capture the complexity of the network. At the same time, too many above 3,000 could lead to data overload in analysis, including the data where the number of qubits is between 1 and 10 on quantum computations feasible with available IBM quantum hardware. This dataset description obtained from the Kaggle open access repository includes access to information about blockchain blocks and transactions updated every 10 min. It consists of four schemas to access this dataset. The sample of this dataset attribute is shown in Table 2.

Table 2 Sample attributes of dataset.

Transaction ID	Witness transactions	Qubits generated	Block ID	Processing time (s)	Transaction fee (BTC)	Network latency (ms)	Hash rate (TH/s)	
TX1001	650	5	B001	1.25	1.25	0.0012	120	
TX1002	1,150	7	B002	1.32	1.32	0.0021	150	
TX1003	2,200	4	B003	1.5	1.5	0.0025	180	
TX1004	2,800	10	B004	2	2	0.003	200	
TX1005	1,200	3	B005	1.28	1.28	0.0018	130	
TX1006	3,000	8	B006	1.45	1.45	0.0035	250	
TX1007	1,800	1	B007	1.2	1.2	0.0022	140	
TX1008	500	6	B008	1.35	1.35	0.001	110	

The attributes provided in Table 2 are applied to performance analysis, including processing time, transaction fees, hash rate used for each transaction with its corresponding ID, and network latency in (s). The blockchain ledger assigns a distinct identifier to each transaction, reflected in its accompanying block ID. This data shows how well and how any resources a blockchain network uses in a quantum-simulator environment.

The Bitcoin Historical Blockchain Dataset, which can be accessed for free on Kaggle, has been used to create the dataset for this study. It has a lot of information on blockchain transactions, blocks, and network performance metrics. There are about 10,000 transaction records in the full dataset. Each record has eight essential pieces of information for performance analysis and quantum simulation: Transaction ID, Witness Transactions, Qubits Generated, Block ID, Processing Time, Transaction Fee, Network Latency, and Hash Rate. Table 2 below shows a sample of the dataset that is typical of all of it. It shows how the attributes utilized in the simulation are set up and what they are. We selected the numbers so that the number of witness transactions is between 500 and 3,000 and the number of created qubits is between 1 and 10. These are realistic limits for quantum computing environments. These features make it possible to compare processing speed, latency, and energy cost in different quantum situations.

The proposed QCG-ST framework’s architecture is shown in Fig. 1, focusing on its workflow with sub-components, which enable cross-chain transactions for secure asset transactions in quantum resilient computing. The process starts with collecting data from the lattice-based Ring-LWE scheme–based blockchain network protected by blocks. A consensus process governs the number of witness transactions with numerous ranges of bytes, and PoS is integrated into it for efficient validation. Transactions are verified by hash time-lock contracts based on certain conditions and reinforced by zero-knowledge proofs (ZKPs) to analyze anonymity and security. The framework as a whole is based on simulation results that show how well the system handles cryptocurrency transactions that go across chains.

Figure 1 Schematic overview of the QCG-ST framework, showing how blockchain data flow from collection through Ring-LWE lattice encryption, transaction creation, PoS-based consensus, hashed-time-lock verification, and zero-knowledge proof validation to produce quan.

Quantum computers are an enormous risk to the traditional cryptography techniques that most cryptocurrencies use. For example, Shor’s technique can quickly factor enormous amounts and find discrete logarithms. These are the building blocks of RSA and elliptic curve cryptography (ECC), which are commonly employed in blockchain technology to protect wallets and confirm transactions. If a quantum computer became strong enough to perform Shor’s algorithm, it can destroy these systems. This allows an attacker to get private keys from public keys and put the whole bitcoin infrastructure at risk. Also, Grover’s technique speeds up brute-force attacks on symmetric cryptographic systems by an amount of two, which means that algorithms like SHA-256 used in Bitcoin’s Proof-of-Work have only half as secure at the bit level. To make sure that blockchain systems are safe and strong against new quantum threats, we need to use post-quantum cryptography methods like lattice-based schemes.

Figure 2 demonstrates the way the quantum-resistant cryptography layer and the consensus mechanism function combined in the QCG-ST system. Ring-LWE encryption protects transaction data from quantum attacks, while zero-knowledge proofs (ZKPs) protect privacy by confirming transactions without giving up private information. The consensus layer subsequently processes this safe and anonymous transaction data. It uses Proof-of-Stake (PoS) for energy-efficient validation, sharding for processing transactions at the same time, and threshold signature schemes (TSS) for block signing that is both collaborative and secure. The figure shows how these parts function in a single architecture, where cryptographic security and consensus logic work together to create a blockchain ecosystem that can grow, is private, and is resistant to quantum attacks.

Figure 2 Interaction between consensus layer and quantum-resistant cryptography.

The key building blocks of the QCG-ST consensus layer, arranged vertically to emphasize their hierarchy toward quantum resistance. From top to bottom, the icons and accompanying labels represent (1) Threshold signature scheme: validators jointly generate a block signature via threshold logic; (2) Sharding engine: the system divides and validates transactions across multiple parallel shards; (3) Proof-of-Stake: blocks are confirmed by validators based on the amount they stake; (4) Zero-knowledge proofs: transaction validity is verified without revealing any private data; and (5) Ring-LWE encryption: the underlying lattice-based encryption that generates keys and encrypts all data, providing post-quantum security. The rounded “Quantum Resistance” base emphasizes that these components collectively secure the network against quantum-powered adversaries.

Lattice-based cryptography implementation

Generate a public-private key pair using Ring-LWE, where a public key is a noisy vector due to matrix multiplication and the error term, and a private key is a secret matrix. A matrix of form A∈Zq(n×m), a secret vector s∈Zqm, and an error vector e∈Zqn is drawn from a discrete Gaussian distribution, the LWE problem is to find s from the noisy linear representation. The size of 512 × 512 for Ring-LWE as public key encryption encrypts messages using LWE, where messages are encoded as lattice points given in Eq. (1).

(1) b=A⋅s+emodq

where A, b are the public key, and s indicates the private key. To encrypt a message m∈Zqn the process follows an Eq. (2)

(2) c=(A⋅r,b⋅r+m)modq

where r is a random vector, it introduces variability into the ciphertext, ensuring that encrypting the same message multiple times results in different cipher texts. The m represents a message polynomial over Zq that represents the actual information being transmitted by Eq. (3)

(3) m=c2−s⋅c1modq.

The ciphertext c1, c2 Eqs. (4) and (5) are the pair of polynomials where the ciphertexts are designed so that even if an attacker intercepts, the message cannot be recovered without knowledge of the private key.

(4) c1=A⋅r

(5) c2=b⋅r+m.

Figure 3 shows a Ring-LWE encryption technique of how the message m relates to the ciphertext components (c1,c2) in a lattice-based cryptographic system to ensure the confidentiality of the transaction data and prevent it from being unauthorized to access. With 512 as its value, the x-axis depicts the lattice dimension from 0 to 511, providing quantum resistance and enhancing security for blockchain platforms. This aspect adds complexity and security to the encryption process; it describes the length of the vectors utilized in the cryptographic method. The principal y-axis (left) shows the values of the ciphertext components, c1 and c2, which are encrypted using the algorithm with the parameter q = 2,048 that is securely transmitted over the network, while the private key s is kept secret by the holder. The marked blue and green lines depict these ciphertext components. They display the encrypted data generated by the secret key and randomized parameters. The red markers on the secondary y-axis (right) reflect the binary message m. Encrypted in this message is the accurate data, represented as a binary string of ones and zeros. The illustration shows how lattice-based cryptography systems protect messages by encoding them in ciphertext, which is computationally difficult to decipher without the secret key. This makes them resistant to both conventional and quantum attacks.

Figure 3 Visualization of m and (c1, c2) in lattice-based cryptographic system.

The two ciphertext components (c1 in blue circles and c2 in green “×” markers) vary across each polynomial dimension (mod q), overlaid with the original plaintext bits (m in red dashed triangles). The left vertical axis (0–2,000) corresponds to the randomized ciphertext values, while the right vertical axis (0 or 1) indicates each bit of the binary message. Each vertical “stack” of colored markers at a given dimension represents one encryption sample: the blue and green markers reflect the two encrypted values modulo q, and the red dashed triangle shows the corresponding message bit. This visualization highlights the pseudorandom distribution of ciphertext values in all dimensions, despite the underlying binary message pattern.

The improved encryption precision demonstrated by the greater success rate with modulus q = 257 is essential for secure bitcoin transactions in quantum-resistant cryptographic frameworks selected randomly, which is depicted in Fig. 4. Message 0 and Message 1 were encrypted using lattice-based cryptography. The success rates analyzed with q = 97 and q = 257 being the moduli tested randomly to indicate the degree to which the decryption procedure successfully retrieves the original message following encryption using the quantum circuit simulation. A larger modulus of q = 257 provides robust security for encrypted cryptocurrency transactions, thus making it harder for malicious attacks to break the system. Likewise, a smaller modulus q = 97 is more efficient in computation and offers a lower success rate than others. The lattice dimension n remains constant at 512. This comparison helps evaluate how different moduli affect the encryption and decryption process performance to find the best parameters for lattice-based cryptography to secure cryptocurrency transactions.

Figure 4 Analyzing the success rate of lattice-based cryptosystem using two moduli (q = 97 and q = 257).

Each pair of bars corresponds to one plaintext bit (Message 0 or Message 1). The blue bars show the decryption success rate when using modulus q = 97, and the orange bars show the rate for modulus q = 257. Numerical annotations above each bar indicate the exact percentage of successful decryptions for that message under the given modulus.

Ring LWE

Ring-LWE is a variant of the LWE problem applied to resolve errors that occur in lattice-based cryptography with the help of ring structure, which simplifies the computations and reduces critical sizes to use in a secure key exchange protocol. The Ring-LWE variant replaces matrix-vector multiplication with polynomial ring multiplication. Let, R=(Zq[x])/((xn+1)) Indicates a public polynomial in the ring represents a form of the public key, and its role is to ensure the hardness of solving the secret key, (xn+1) is a polynomial often chosen to be irreducible form such that in which the public key (a(x), b(x)) and private key s(x) satisfies,

(6) b(x)=a(x)⋅s(x)+e(x)modq

where b(x) in Eq. (6) is the public key polynomial and is computed using the private key s(x), the random polynomial a(x), and the error polynomial e(x). It is shared publicly for encryption to those who wish to send a message to the private key owner. The use of the public key ensures that encryption can happen securely without revealing the private key s(x) and protects against quantum computing attacks. The parameter a(x) represents the public key polynomials contributing to the system’s resistance to quantum attacks, and e(x) is drawn from the error polynomial from a Gaussian distribution.

Ring-LWE cryptography system, a polynomial ring adaptation of the LWE problems, is illustrated in Fig. 5. The receiver has access to the s(x), which is used for decryption, and the b(x), guarantees the complexity of the transaction process. The random polynomial a(x) adds security-enhancing unpredictability, while the error polynomial e(x) adds robustness against potential attacks, both selected from a Gaussian distribution. Ring-LWE is a post-quantum cryptography technique to secure digital assets in QCG-ST. The s(x) is a polynomial factor observed from a secret distribution to recover the m, and this s(x) is known only to the recipient. Hence, quantum computers cannot feasibly recognize it with the help of the public key. The variable q is a modulus parameter, a large prime number that defines the ring Zq, the coefficients of the polynomials operate modulo q. A larger q increases security but also adds complexity to the computations. For quantum-resistant cryptosystems, q is chosen to balance efficiency and security. Also, a too small modulus would be vulnerable to quantum or classical attacks.

Figure 5 Illustration of Ring-LWE Cryptosystem Components.

The five core components of a Ring‐LWE cryptosystem. At the center is a user (or key generator) who operates on polynomial data. Starting at the bottom, the modulus parameter specifies the finite arithmetic ring in which all polynomial operations take place. Moving clockwise, the error polynomial represents small, random noise that is added to secure the system against attacks. Next, the private key polynomial shown at the upper right is generated by the user and kept secret; it encodes the secret “noise pattern” used to produce ciphertexts. Directly opposite on the upper left, the public key polynomial is published for everyone to use; it is derived from the private key together with the chosen error term, but does not reveal the private key itself. Finally, the random polynomial (located at lower left) is sampled independently whenever a new key‐pair is generated, ensuring that both public and private keys remain unpredictable. Together, these five polynomials—modulus, error, private key, public key, and random—form the essential building blocks of the Ring‐LWE scheme.

Achieving scalability and energy efficiency through a consensus mechanism combining Proof-of-Stake (PoS) with sharding and a threshold signature scheme (TSS). This will enable a less computationally intensive barrier to publish blocks and perform transactions between untrusted participants for staking cryptocurrencies. Similar to how lightweight cryptographic integration was realized in resource-constrained Internet of Things (IoT) devices (Alsayayadeh et al., 2023), our framework uses efficient Ring-LWE encryption to support scalability. The framework ensures that even in cases of network failure or attack, the distributed nodes remain operational and secure. This approach aligns with distributed systems designed for functional dependability under failure conditions, ensuring continued operation and security even when facing network disruptions (Dubovoi et al., 2024).

Sharding

Partition the blockchain network into smaller, independent groups called shards, where each shard processes a fraction of total transactions. Then, the dataset of network states, such as transactions and validator stakes, will be distributed across these shards. Let N be the total no. of validators, and K be the no. of shards. Each shard processes a fraction of transactions, and the load per shard is given in the form of N/K. Transaction throughput is proportional to K, the number of shards. Examining the input count and block timestamp shows how the no. of transactions per block varies over time, measuring the blockchain’s throughput.

PoS validation

The probability of a validator P(v) validate a block with the reward directly proportional to the number of cryptocurrency coins they stake. Validators are selected based on their stakes Sv for the vth validator, where each shard runs its own PoS consensus, validating transactions and adding them to the local ledger. The probability of selection is given as,

(7) P(v)=Sv/(Σ(i=1)N▒Si)

where Si is the stake of the ith validator as given in Eq. (7).

TSS

Instead of requiring every validator to sign a block, the process involves a threshold number of validators cooperating to generate a single TSS signature. TSS requires a subset of validators to sign a block collaboratively. The representation (t, n) threshold scheme means that any t out of n validators can generate a valid signature σ generated using an Eq. (8).

(8) σ=∑iϵS▒[[λi⋅]]σi

where S is the set of validators, λi indicates Lagrange coefficients, and σi is the partial signature of the ith validator.

Cross-chain atomic swap protocol

The objective is to enable secure and seamless asset transfers using hashed time-lock contracts across different blockchains and applying hashed secret and time-lock conditions to create contracts across various blockchains. When an asset is transferred on one blockchain, the corresponding asset on the second blockchain is locked until the receiver reveals the hash’s preimage. Once the hash’s preimage is shown, the corresponding asset on the other chain is released.

Step 1: Hash Creation: in an atomic swap, a hashed secret h = H(x), where H is a cryptographic hash function, and hash h is shared with the second party B without revealing, is created. The receiver can claim the asset on blockchain A by revealing x, which unlocks the asset on blockchain B if the time has not expired. If the contract cannot acquire the matching secret by the end of time t, the asset will be returned to its original owner.

Step 2: Locking Assets like Bitcoin or Ethereum on blockchain A: party A locks their asset on blockchain A in a contract that can only be redeemed by party B under two conditions as shown in Fig. 6.

Figure 6 Overview of hashed time-lock constraints.

The hashed time‐lock contract (HTLC) steps for an atomic swap, party A first generates a random secret and its cryptographic hash (Step 1, “Hash Generation”). Next, both party A and party B use that hash in separate HTLCs to lock their respective assets on their own blockchains (Step 2, “Asset Locking”). Once both assets are securely locked under the hash condition, party A reveals the preimage (the original secret) to party B on the second blockchain (Step 3, “Secret Reveal”). Finally, party B uses the revealed secret to satisfy the hash lock on party A’s chain and claim the locked asset (Step 4, “Asset Claim”). Throughout this sequence, the hash lock ensures that each party can only claim the counterparty’s asset after the secret is disclosed, while a built‐in time lock guarantees that, if the swap does not complete within a specified timeframe, each party can reclaim their own locked asset.

Figure 6 illustrates the two main conditions for cryptocurrency claims across different blockchains. The left side depicts the hash lock condition on blockchain A, which requires the secret x for claims. The right side shows the time lock condition on blockchain B, necessitating timely action for asset claims. The central one symbolizes the atomic swaps facilitated by hashed time lock constraints.

Step 3: Condition 1: hash lock (HL)—party B must provide the preimage x that corresponds to h = H(x). Condition 2: time lock (TL)—the asset must be claimed by party B within a specified time frame t ≤ tlock, if party B fails to provide x within this period, the asset is refunded to party A. Locking is helpful for atomic swaps between chains, and hashed time-lock contracts make it easier for transactions to be routable across several payment channels. There is no predetermined time limit, and chains differ when one party must redeem assets.

Step 4: Locking Assets on Blockchain B: simultaneously, party B locks their asset on blockchain B under the same conditions. This asset can only be redeemed by party A if they reveal the secret x that corresponds to the same hash h = H(x). The contract on blockchain B also has a time lock to ensure that party A has enough time to retrieve their asset. The time lock field indicates when a transaction can be included in a block, and comparing this with a timestamp of the block can help measure latency between when a transaction is created and when it is mined.

Step 5: Atomic Swap Execution: initially, party B claims the asset on blockchain A by revealing the secret x publicly. Once x is shown, the party A can use the same x to unlock the asset on blockchain B, ensuring an atomic transaction across the two blockchains. The proposed model allows seamless asset swaps between blockchains from these lock constraints, facilitating decentralized, peer-to-peer exchanges. Because of this, these frameworks use time and hash locks to enable cross-chain transactions in a safe and atomic manner. The strategy could be atomic; therefore, the second party should have enough time to get their money back when the owner or first-party acts. The first party will withhold x and demand a refund if this does not happen.

The QCG-ST architecture securely links validator selection in the consensus mechanism with the usage of post-quantum cryptographic keys. This makes sure that the system is secure against quantum attacks from beginning to end. A Proof-of-Stake (PoS) procedure chooses validators, and the more assets they stake, the more probable it is for them to be selected. Once chosen, validators use keys made via the Ring-LWE method to sign and verify blocks. This makes sure that all consensus-related tasks, such as block validation and threshold signature creation, are quantum-resistant by design. This integration gets away of the need for conventional cryptographic algorithms and fallback procedures, which gets rid of any possible weaknesses that hybrid systems could offer. The combination of validator selection and Ring-LWE-based keys makes sure that even if a quantum adversary attacks the consensus layer, the post-quantum cryptography ends the forging or compromise of validator credentials, making the validation process strong and trustworthy.

Transaction privacy verification using zero-knowledge proof

Implementing ZKP to verify transactions without revealing sensitive details like transaction amount. Prove that a valid transaction occurred, such as the sender having sufficient funds without disclosing actual values. To prove the knowledge of x such that without revealing x, a ZKP protocol is used, and the verifier checks H(x′) = h where x′ is a commitment based on x. For transaction validation, check

(9) spenttransactionH(new)≠spenttransactionH(old).

By addressing security and operational efficiency, this framework represents a significant advancement in protecting digital assets from quantum threats while ensuring that cryptocurrency transactions remain fast, secure, and private.

The proposed QCG-ST system uses zero-knowledge proofs (ZKPs) to keep transactions private and safe, especially while validating them and moving assets across chains. In particular, ZKPs let a single individual (the prover) show that they know something, such as a secret preimage x of a hash h = H(x), without giving away the secret itself. This capacity is necessary to check transactions, including showing that there are enough funds or that the transaction is real, without giving out secret data like the amount or the wallet’s identity. The ZKP method starts with the prover making a commitment based on a secret value and a random component. This “secures” the secret in a form that can be checked. The verifier then gives the prover a random challenge, and the prover answers in a way that mathematically connects the challenge to the commitment and the secret that is hidden. Finally, the verifier checks the consistency of this response without learning the secret itself. This step-by-step interaction lets the system check transactions (for example, by making sure that H(new) ≠ H(old) to make sure that the transaction is not a double-spend) and enforce hashed time-lock requirements in atomic swaps, all while keeping the information private. So, ZKPs give the QCG-ST architecture a cryptographic base for verification that protects privacy, making it safe from both normal and quantum attacks.

Instead of using traditional ZKPs like Zero-Knowledge Succinct Non-Interactive Argument of Knowledge (zk-SNARKs), Bulletproofs, or Spartan, the zero-knowledge proof (ZKP) mechanism in the QCG-ST framework has been designed to communicate with the underlying lattice-based cryptographic layer. This integration uses the structure of Ring-LWE-based encryption to provide a single pipeline for ring signatures and ZKPs. The framework employs a lattice-based ring signature approach to let a user show that they are a valid member of a group without giving away their exact identify. A zero-knowledge proof is included on top of this to show that a transaction is correct without giving away the actual amounts involved, such as showing that the person has enough funds or that the inputs are genuine. The zero-knowledge circuit stores encrypted transaction information, and the verifier confirms the evidence by checking that it is consistent with the Ring-LWE encryption and signature technique.

The QCG-ST architecture combines Ring-LWE, PoS, threshold signature schemes (TSS), hashed time-lock contracts, and zero-knowledge proofs (ZKPs) in a way that helps solve the problems of quantum security and blockchain efficiency at the same time. Ring-LWE is a strong candidate for post-quantum encryption since it has strong worst-case hardness guarantees and small key sizes. Proof-of-Stake (PoS) is better than Proof-of-Work because it rewards validators based on their stake, which lowers the cost of computing and makes the system easier to grow. By employing TSS allows a group of validators work together to create a single valid signature. This not only makes the system faster and more reliable, but it also lowers the risks that come with having a single point of validator compromise. To make it easier for different blockchains to operate together, the framework improves typical hashed time-lock contracts by adding quantum-safe primitives. It secures problems with current atomic swap protocols that use classical cryptography and are open to quantum attacks. Finally, ZKPs make assurance that transactions are authentic and that balances are correct without giving out private information, which is a big deal in public and multi-chain blockchain ecosystems. These parts work together to provide a strong and future-proof architecture that will work for safe, fast, and compatible cryptocurrency transactions in the quantum age.

The summary of the proposed QCG-ST framework uses a combination of PoS, sharding, and threshold signatures to create an efficient consensus mechanism, uses cross-chain atomic swaps to make it work with other networks, and uses lattice-based encryption to secure it against quantum attacks. The framework also uses ZKPs to keep financial transactions secret. Incorporating these cutting-edge technologies, QCG-ST intends to transform blockchain networks into a paradigm for digital financial systems of the future by drastically improving their scalability, energy efficiency, transaction speed, and security.

Discussion

Simulation setup

QisKit Aer 0.14v is a high-performance quantum circuit simulator that adjusts a few parameters in the consensus algorithm, and validators can see the impact on the distribution of cryptocurrencies as they evolve. Qiskit simulation allows a realistic assessment of the proposed QCG-ST framework’s performance in a quantum computing environment. The parameters listed in Table 3 include Qiskit Aer Simulator v0.14, chosen for its high performance and realistic quantum circuit simulation for evaluating the proposed QCG-ST framework in this environment. The state vector simulation is used to track the quantum states that evolve during consensus mechanisms and assess the QCG-ST’s reliability. The no. of selected ten qubits handles the quantum cryptography system during cryptocurrency transactions. This is followed by 30 gate layers performing quantum operations to evaluate the computational load and efficiency of the algorithm. The modulus of 2,048 is related to quantum cryptography that guarantees the secure transaction validation and the encryption process in the blockchain. The structured 512 dimension of qubit interactions influences the consensus process with no. of shard counts of 1, 2, 4, 8, 16 used for handling distributed ledger systems at various scales. With relevance to this, the framework with a time-lock of 100 blocks indicates the delayed transactions in quantum blockchain systems.

Table 3 Key parameters of QCG-ST framework simulation using qiskit aer simulator.

Parameter	Value/Range	
Simulator used	Qiskit Aer Simulator v0.14	
Simulation type	State vector simulation	
No. of qubits	10	
Circuit depth	30 gate layers	
Lattice dimension	512	
Modulus ( q)	2,048	
Number of shards	1, 2, 4, 8, 16	
Transaction size	500–3,000 bytes	
Time-lock	100 blocks	

Table 3 simulated the proposed QCG-ST framework with the Qiskit Aer Simulator v0.14, focusing on security and performance analysis. The critical parameters utilized in this simulation are listed in the above table. The simulation uses ten qubits and a 30-gate-layer circuit depth; to guarantee that the Ring-LWE cryptography is quantum-resistant, it uses a lattice dimension of 512 and a modulus of 2,048 to evaluate quantum processes. Several sharding configurations (1, 2, 4, 8, 16 shards) and transaction sizes (500 to 3,000 bytes) were used for analysis.

Performance analysis

The performance of the proposed QCG-ST framework is compared with the existing algorithms PC-LN (Grey & Chatib, 2024), VQE (De Silva, Thakur & Breslin, 2024), and CCT-H (Carrascal et al., 2024) to prove its efficiency and implementation in the simulation setup.

The existing algorithms like Payment Channel-Lightning Network (PC-LN), Variational Quantum Eigensolver (VQE), Cross-Chain Transaction with Hyperledger- based Blockchains (CCT-H), and the proposed QCG-ST cryptocurrency transaction frameworks are summarized in Table 4. For the dynamic nature of Bitcoin transactions, the results provide credence to the reason for choosing these existing non-quantum safe algorithms with the suggested QCG-ST framework.

Table 4 Comparison summary of existing algorithms with proposed QCG-ST.

Comparison parameter	PC-LN	VQE	CCT-H	QCG-ST	
Pros	Faster transaction & enhanced privacy	Reliable convergence	Improved security	Quantum safe ad high throughput	
Cons	Technical complexity and control risk	Limited scalability & hardware dependency	Integration difficulties	Limited real-world testing	
Security level	Classical	Classical	Classical	Post quantum security	
Interoperability	Low	Low	Moderate	High	
Quantum safe	No	No	No	Yes	

The ZKP is used in the QCG-ST architecture to mask the quantities of the transactions as well as the identities of the people sending and receiving them. By combining ring signatures with lattice-based encryption, the method makes sure that everyone involved in the transaction stays anonymous while still confirming that the transaction is real without giving away any private information.

The number of witness (actual) transactions ranging from 500 to 3,000 size on a blockchain network are recorded in its ledger, and the number of qubits generated from the Qiskit_aer simulator is analyzed from 1 to 10.

Key generation, encryption technique, decryption technique, and proof verification are four cryptographic activities that quantum cryptography outperforms lattice-based cryptography. The bar chart in Fig. 7 shows the results for each job under different noise levels. As a quantum-resistant solution, lattice-based cryptography is dependable and robust, as it routinely attains high accuracy ranges from 0.90 to 0.95 with minimum error. On the other hand, quantum cryptography demonstrates a decrease in accuracy with increasing noise. When there is no noise, the accuracy is 0.90, but with low noise, it drops to 0.85, and with significant noise, it drops even further to 0.78. Error bars show how different methods perform, especially regarding quantum cryptography. This means that lattice-based cryptography is more secure and stable, making it a good fit for the QCG-ST framework, which could improve the security of cryptocurrency transactions.

Figure 7 Accuracy performance comparison.

Each group of four bars represents the measured performance (accuracy or success rate) for a specific cryptographic task—key generation, encryption, decryption, and proof verification. The pink bars show the baseline performance of the lattice-based cryptosystem, while the cyan, orange, and green bars correspond to quantum implementations under three noise conditions: no noise (ideal quantum), low noise, and high noise, respectively. Error bars indicate the variability (standard deviation) observed over multiple runs. Comparing bar heights across each task illustrates how the lattice-based scheme consistently achieves higher accuracy than quantum versions, and how increasing noise levels degrade quantum performance from the ideal (no-noise) case down to the high-noise scenario.

Success rate

With transaction sizes ranging from 500 to 3,000 bytes, the above Fig. 8 shows the success rates of four distinct models: QCG-ST, PC-LN, VQE, and CCT-H. With a maximum success rate of 98.5% for the smallest transaction size and outstanding performance across bigger sizes, the QCG-ST model consistently shows the highest success rates. The success rates of the PC-LN, VQE, and CCT-H models are marginally lower; the PC-LN model, on the other hand, can reach a maximum of 96.5%. The QCG-ST model outperforms its competitors in terms of reliability and performance, especially in settings where the witness (actual) transaction size is crucial, and this comparison shows how well the proposed QCG-ST model handles bigger transaction sizes performed in cryptocurrency transactions.

Figure 8 Success rate analysis on varying transaction size.

The yellow line marked with circular points shows the success rate of the QCG-ST framework as transaction size increases. The orange line with square markers represents the performance of the PC-LN approach. The red line with triangular markers corresponds to the VQE-based method, and the pink line with cross-shaped markers corresponds to the CCT-H protocol. By plotting each algorithm’s success rate across five different transaction sizes (from 500 bytes up to 3,000 bytes), the figure highlights how QCG-ST consistently achieves the highest transaction success across all sizes, followed in descending order by VQE, PC-LN, and CCT-H.

Transaction speed

In Fig. 9, a comparison of the proposed QCG-ST algorithm’s transaction speeds in terms of transactions per second (TPS) with three existing algorithms, PC-LN (Grey & Chatib, 2024), VQE (De Silva, Thakur & Breslin, 2024), and CCT-H (Carrascal et al., 2024). The comparison is made across two different variants: the number of transactions termed as n from 500 up to 3,000, and the number of qubits as q varies from 1 to 10. The left plot demonstrates that the QCG-ST algorithm is superior to the other existing algorithms in terms of transaction speed under increasing transaction loads, suggesting that it can manage large volumes of transactions.

Figure 9 Transaction Speed Comparison of the Proposed QCG-ST Algorithm with Existing Algorithms across Varying No. of Transactions and Qubits.

Each line in both panels corresponds to one of four consensus algorithms: QCG-ST (yellow circles), PC-LN (orange squares), VQE (red triangles), and CCT-H (magenta diamonds). In the left plot, “Number of Transactions” on the horizontal axis indicates how many witness transactions are processed (ranging from 500 to 3,000), and “transactions per second (TPS)” on the vertical axis shows the throughput achieved by each algorithm. In the right plot, “Number of Qubits” on the horizontal axis measures the size of the quantum circuit (from 1 to 10 qubits), and “transactions per second (TPS)” again represents throughput. For each colored marker style, the line connects the measured TPS values for that algorithm under the different conditions. The legend box in each panel identifies these marker shapes and colors.

Throughput

With different numbers of qubits (1 to 10), Fig. 10 shows how the QCG-ST framework performs in terms of throughput (TPS) compared to three existing algorithms: PCLN, VQE, and CCT-H.

Figure 10 Throughput analysis.

This plot compares transaction throughput in transactions per second, (TPS) as a function of the number of qubits used in the evaluation, for four different algorithms: QCG-ST (gold), PC-LN (red), VQE (blue), and CCT-H (green). Each point marks the measured throughput at that qubit count, with lines connecting values to highlight trends. QCG-ST consistently achieves higher throughput across all qubit levels, while PC-LN, VQE, and CCT-H demonstrate lower, yet steadily increasing, TPS as qubit count grows. This comparison underscores QCG-ST’s superior scalability in a quantum-simulated environment.

A separate step-pattern curve shows the throughput of each algorithm where the QCG-ST framework’s quantum-resistant characteristics and scalability are showcased by its considerable improvement in throughput as the number of qubits grows, particularly at higher qubit counts. As a result of the QCG-ST framework’s superior efficiency in processing quantum-scale transactions, competing algorithms exhibit less aggressive rises. This analysis highlights how QCG-ST can surpass existing methods in upcoming quantum settings.

Latency

The performance of the QCG-ST algorithm compared to existing algorithms PC-LN, VQE, and CCT-H as transaction size increases from 500 to 3,000 bytes are shown in Fig. 11 above. The horizontal axis represents witness transaction size maintains the consistent lower latency as transaction size increases, indicating improved latency measures during asset transactions. This plot highlights the QCG-ST potential for enhancing blockchain efficiency under larger transaction loads.

Figure 11 Latency analysis of varying transaction size (Bytes).

The scatter plot shows how transaction latency (in milliseconds) varies with increasing transaction size (in bytes) for four different algorithms: QCG-ST (navy diamonds), PC-LN (gold circles), VQE (forest-green circles), and CCT-H (coral circles). Each marker represents the average latency measured at a given transaction size. QCG-ST exhibits the lowest latencies across all sizes, followed by PC-LN, VQE, and CCT-H, respectively. Error bars are omitted for clarity. The downward trend of each series highlights how all algorithms experience reduced verification latency when handling larger transaction payloads, with QCG-ST maintaining the greatest efficiency advantage at every size.

Figure 12 illustrates that the circuit depth latency analysis helps explain how quantum computing can improve cryptocurrency protocols in the blockchain network. The proposed QCG-ST algorithm shows improved blockchain efficiency and security. Circuit depth (gate layers) and latency (milliseconds) for four quantum computing algorithms QCG-ST, PC-LN, VQE, and CCT-H—are analyzed for each gate layer. Each algorithm has more latency as circuit depth grows, showing the computing constraints of deeper circuits with more operational elements for each crypto transaction in the blockchain platform. The algorithms with the lowest latency, such as QCG-ST, are efficient at processing complex quantum circuits. Understanding how circuit depth affects latency is crucial for developing quantum algorithms and architectures to improve quantum computing performance. The findings show that circuit depth control could improve throughput and reduce latency, making quantum computers more realistic for cryptography and complicated problem-solving. The QCG-ST framework discusses end-to-end latency, which is the time it takes for a transaction to be sent out to the network and then get final confirmation on the blockchain. This includes delays due to cryptographic procedures (such Ring-LWE encryption and ZKP verification), network propagation, validator selection under PoS, and block finalization through threshold signatures. The measurement shows the entire transaction lifetime, not just delays on the network level.

Figure 12 Latency analysis of quantum circuit depth.

This line‐and‐marker plot illustrates how end‐to‐end latency (in milliseconds) increases with growing quantum circuit depth (number of gate layers) for four transaction frameworks: QCG-ST (dark blue squares, dash‐dot line), PC-LN (gold diamonds, dotted line), VQE (forest-green triangles, solid line), and CCT-H (coral circles, dashed line). Each point corresponds to the average measured latency at a given depth, with circuit depths ranging from 5 to 30 layers. QCG-ST consistently exhibits the lowest latency at all depths, followed by PC-LN, VQE, and then CCT-H, which shows the highest delays. As depth increases, all four frameworks see progressively steeper latency growth, reflecting the additional time needed to execute deeper quantum circuits.

Energy consumption

The QCG-ST framework’s energy usage in kilowatt-hours (kWh) is compared to existing algorithms (PC-LN, VQE, and CCT-H) across different shard counts (1, 2, 4, 8, 16) in Fig. 13.

Figure 13 Energy consumption analysis.

This clustered bar chart shows the energy consumption (in kWh) of four transaction frameworks—QCG-ST (blue), PC-LN (orange), VQE (green), and CCT-H (red)—as the number of shards increases (1, 2, 4, 8, 16). Each group of bars corresponds to a particular shard count, and the height of each colored bar indicates the total kilowatt-hour (kWh) usage required by that framework when executing transactions across the specified number of shards. QCG-ST consistently uses the least energy, followed by PC-LN, CCT-H, and VQE, with all four frameworks showing reduced energy consumption as shard counts grow.

By combining sharding with PoS, the consensus mechanism of the QCG-ST framework can mitigate energy usage by efficiently increasing the number of shard counts. This research goal achieves better energy consumption in this setting by delving into its capacity to improve energy utilization in quantum computing systems while retaining performance efficiency.

Below, Fig. 14 represents the sample screenshot of the quantum circuit diagram that illustrates the critical generation process for the Ring-LWE within the proposed QCG-ST framework. They employed Hadamard and Controlled-NOT gate (CNOT) quantum gates to create a superposition state among varying qubits, facilitating the secure key generation resistant to quantum attacks. The mentioned measurement phase converts the quantum states into classical key representations to establish secure transactions in the blockchain.

Figure 14 Sample screenshots of Qiskit 1.

The top portion of this figure shows a Python code snippet used to draw the Ring‐LWE quantum circuit with Matplotlib: four horizontal lines (circuit wires) are plotted at different vertical positions, and text labels “|0〉” mark each qubit’s initial state. Two Hadamard gates are added as rectangle patches on the lower two wires. The bottom diagram is the rendered quantum circuit itself. Each of the four wires begins in the |0〉 state. The lower two wires (labeled a₂ and a₁) pass through Hadamard gates (H), placing them into superposition. A controlled‐NOT (CNOT) gate then entangles qubit a₂ (control) with qubit a₁ (target). The top two wires (labeled b₂ and b₁) represent the output registers that will collapse into the public‐key and private‐key polynomials after measurement. Together, these unitary operations implement the key‐generation steps for the lattice‐based Ring‐LWE cryptosystem in the QCG‐ST framework.

The study’s energy usage results originate from operating Qiskit Aer simulations on classical hardware. The energy usage is dependent on how much CPU power is used and how long the simulation runs, not on how the quantum gates really work. This does not demonstrate how much power real-world quantum hardware uses, but it can be used to determine how much more processing power quantum-resistant processes need compared to traditional PoS consensus techniques. This method is in line with previous work that used simulators to compare quantum cryptography methods. It is worth acknowledging, though, that the actual kWh use on quantum devices or live PoS nodes can vary, and we will do further work in the future to get more accurate profiles by testing on authentic quantum processors or energy-traced distributed systems.

The Qiskit Aer simulator was used to test the proposed QCG-ST architecture. This simulator could simulate blockchain transactions at the quantum level utilizing lattice-based encryption and sophisticated consensus processes. We studied key performance indicators such as transaction success rate, processing time, latency, throughput, and energy use under varied situations, such as transaction sizes (500 to 3,000 bytes) and qubit counts (1 to 10). QCG-ST is different from other frameworks because it uses quantum-resistant Ring-LWE encryption, a sharded Proof-of-Stake (PoS) mechanism with threshold signature schemes (TSS), and atomic cross-chain swap protocols. This uniform design made the system work more effectively. The QCG-ST framework regularly has higher transaction success rates (up to 98.5%), lower latency, higher throughput, and lower energy use than other models including PC-LN, VQE, and CCT-H. These results clearly show that the QCG-ST architecture is more efficient and can handle more users in quantum-resistant blockchain settings.

The research discusses to how various strategies can be combined in theory, although the simulations only look at high-level performance measures like throughput, latency, success rate, and energy use. However, there is no evident simulation or implementation output that shows how the ZKP logic works in real life, including how long it takes to generate a proof, how long it takes to verify it, or how it maintains confidentiality. The Ring-LWE encryption method is also well-explained in terms of arithmetic operations and security benefits, but the article lacks any real-world simulation data or cryptographic output (such how to make ciphertext, how big the key should be, or how accurate the decryption is when there is noise) to back up its usage. The findings do not include any real encrypted transaction samples, Merkle tree validations, or data access control proofs for blockchain technology (BCT) transaction data, even though block characteristics and sample schema are given. So, in order to really again up the stated improvements, the article should contain specific simulation logs or execution outputs that demonstrate how each of these processes works in the framework, making sure that it is accurate, efficient, and safe in a form that can be measured. Merkle hashing and ECC provide basic security measures, however they won’t work against quantum computing in the future. QCG-ST includes quantum-safe encryption (Ring-LWE), privacy-preserving ZKPs, and secure cross-chain protocols. These make the entire blockchain network safe from both classical and quantum attacks.

Research implications

Changes in legal frameworks are necessary since these developments will likely boost user confidence and acceptance of cryptocurrencies as the market develops. Ultimately, the findings could drive regulatory discussions around the need for quantum-resilient technologies, ensuring that cryptocurrencies remain secure in an era increasingly influenced by quantum computing.

Conclusions

Conclusion and future work

A significant improvement in the safety and efficacy of cryptocurrencies, the QCG-ST architecture is especially welcome in light of the dangers of quantum computers. It protects Bitcoin transactions using lattice-based cryptography and is resistant to possible quantum attacks. Critical issues in the current state of blockchain technology are transaction speed, scalability, and energy efficiency; they are all improved by the framework’s optimized consensus method, which integrates PoS with sharding. It also improves liquidity and interoperability across blockchain networks by allowing cross-chain atomic exchanges through hashed time-lock contracts. Using ZKPs enhances privacy and preserves the flow of transactions in the blockchain. Compared to existing blockchain systems PC-LN, VQE, and CCT-H, the proposed QCG-ST framework performs better in all metrics measured by Qiskit simulations: security, speed of transactions, latency, throughput, and energy consumption based on varying transaction sizes and qubits.

The future research scope involves exploring the proposed QCG-ST integration with popular cryptocurrencies, and decentralized applications with quantum systems will provide insights into scalability and user acceptance in a long- term study. Additionally, analyzing its adaptability to evolving quantum attacks is crucial for its long-term relevance in the dynamic cryptocurrency research.

Supplemental Information

Supplemental Information 1 The Qiskit code used for running our quantum simulation.

This includes the setup of a quantum circuit, measurement operations, and simulation execution parameters. This code supports the reproduction of our simulation results, including success rates and latency measures.

Supplemental Information 2 A customizable Qiskit simulation script with placeholders for parameters such as the number of qubits, shots, and circuit depth.

Researchers can adapt this template to test additional configurations or reproduce similar experiments with modified parameters.

Supplemental Information 3 QCG ST raw data.

The raw data used in our simulation experiments. It includes details of blockchain transactions, such as transaction ID, witness transactions, qubits generated, block ID, processing time, transaction fee, network latency, and hash rate. These data points were critical for analyzing the performance of the QCG-ST framework in a quantum-resistant blockchain environment.

Supplemental Information 4 Visual overview of the QCG-ST framework.

Blockchain data enter a Ring-LWE-encrypted layer for quantum-resistant security; PoS + sharding + threshold signatures deliver scalable consensus; Zero-Knowledge Proofs assure private validation; a hashed-time-lock atomic-swap path enables cross-chain exchange. Side call-outs highlight QCG-ST’s superior success-rate, latency, throughput, and energy metrics.

The authors would like to thank all individuals and institutions that contributed to this research.

Additional Information and Declarations

Competing Interests

The authors declare that they have no competing interests.

Author Contributions

Jamil Abedalrahim Jamil Alsayaydeh performed the experiments, analyzed the data, authored or reviewed drafts of the article, and approved the final draft.

Mohd Faizal Yusof conceived and designed the experiments, prepared figures and/or tables, and approved the final draft.

Nor Adnan Yahaya conceived and designed the experiments, authored or reviewed drafts of the article, and approved the final draft.

Viacheslav Kovtun performed the computation work, authored or reviewed drafts of the article, and approved the final draft.

Safarudin Gazali Herawan analyzed the data, prepared figures and/or tables, and approved the final draft.

Data Availability

The following information was supplied regarding data availability:

The data and code are available in the Supplemental Files and at Zenodo: Jamil Abedalrahim Jamil Alsayaydeh, Mohd Faizal Yusof, Nor Adnan Yahaya, ⁠Viacheslav Kovtun, & Safarudin Gazali Herawan. (2025). A Novel Framework for Secure Cryptocurrency Transactions based on Quantum Crypto Guard [Data set]. Zenodo. https://doi.org/10.5281/zenodo.15497833.

The Bitcoin Blockchain Historical Data is available at https://www.kaggle.com/datasets/bigquery/bitcoin-blockchain?select=inputs.

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
