# Peer review of "A novel framework for secure cryptocurrency transactions using quantum crypto guard"

_PeerJ Computer Science, doi:10.7717/peerj-cs.3030_

## Round 0.1 · original submission · Major Revisions

· Academic Editor

Major Revisions

Incorporate the suggestions of the reviewers.

Reviewer 1 ·

Basic reporting

The authors proposed a compelling quantum-resistant framework combining PoS, threshold signatures, cross-chain atomic swaps, and lattice-based cryptography. The methodology is timely, and the simulation results using Qiskit Aer are promising. However, the current manuscript needs major revisions to:
1. Include a concise explanation of Shor’s algorithm or Grover’s algorithm to show how quantum computers specifically endanger classical cryptography used in most cryptocurrencies.
2. Emphasize which transaction steps or cryptographic primitives (hashes vs. signatures vs. consensus steps) are the most vulnerable to quantum attacks.
3. Consider citing more recent standardization activities, such as NIST’s post-quantum cryptography competition, to justify the choice of Ring-LWE over other lattice-based approaches (e.g., NTRU, KYBER, SABER). Show how your chosen scheme is relevant and validated by the community.
4. The existing Figure 1 is helpful, but consider adding one additional figure that focuses exclusively on the interplay between the consensus layer (PoS + sharding + threshold signatures) and the quantum-resistant cryptography (Ring-LWE + zero-knowledge proofs).
5. Ring-LWE, PoS, threshold signature scheme (TSS), hashed time-locks, and ZKPs are introduced but not justified. Add details, e.g., “Why TSS for partial signatures?” or “What is the novelty of your cross-chain solution vs. hashed time-lock in existing atomic swap protocols?”.
6. For the new consensus, specify how the selection of validators works in synergy with the quantum-resistant keys. Are validators using post-quantum keys exclusively, or is there a fallback?
7. ZKPs are discussed in broad strokes. If you are leveraging standard ZKPs (e.g., zk-SNARKs, Bulletproofs, Spartan), provide a short overview. If it is a custom scheme, show how it integrates with the lattice-based framework (for example, ring signatures + ZKPs in a single pipeline).

Experimental design

8. Clarify if the ZKP is used only to hide transaction amounts or also to hide the identities of the sender/receiver.
9. The paper discusses latency in a general sense, but is it the end-to-end latency (from transaction broadcast to final confirmation), or just network-level latency? Clarify how you measure it.

Validity of the findings

10. The results for energy usage are promising. However, readers may wonder how you map a Qiskit Aer simulation’s gate count or circuit depth to real-world kWh. Provide a short justification or reference: quantum simulators on classical hardware measure CPU usage differently from actual quantum devices or classical PoS nodes.
11. You show throughput for 1–10 qubits and some number of shards (1, 2, 4, 8, 16). It would be valuable to see if the advantage saturates or continues to grow. For instance, does throughput plateau after 8 shards, or does it keep increasing?
12. If possible, incorporate at least one well-known purely classical competitor (e.g., a standard ECDSA-based PoS approach) to highlight the quantum advantage.

Reviewer 2 ·

Basic reporting

In this work, the authors mentioned that proposed a novel blockchain-based framework, i.e, Quantum Crypto Guard for Secure Transactions (QCG-ST). However, the following comments should be considered for raising the quality of the paper as well as the readers.
1. In the abstract, authors should need to address the methods or protocols or type of blockchain, or data source details. As well as identified research gaps and your proposed work performance briefly.
2. In the introduction section, suggested to revise the contribution terms by including the flow of connection to all the points.
3. Suggested to include a summary table in the literature survey by considering certain qualitative factors.
4. Please check the content format as well as sub-section heading, i.e, Blockchain Speed And Interoperability Issues in the Literature survey, how much this one is suitable for your work, as well as the addressed content in the section.
5. Table 1 shows the sample attributes of the dataset. But should need to address the total dataset details like total no. of records and no. of attributes, etc.
6. The authors did not include the total dataset in the raw data. Are you considering only a 9-record dataset for your proposed framework? How could you justify that your system is efficient if not taken the big dataset?
7. How could the authors evaluate the performance, and how does your framework's efficiency show much better than existing frameworks
8. Lattice-based Cryptography is a broad term; many algorithms fall under that. Which cryptographic algorithm was chosen by Ring -LWE uses to generate the paired keys? Suggested to clearly address the encryption and decryption, and key generation process as algorithm-wise
9. In line number 263, “The size of 512×512 for Ring-LWE as public key encryption encrypts messages using LWE.” What does "size of 512 X 512" mean…Justify
10. So many places, formatting, and language issues were there. Revise the content
11. The authors addressed that the ZK proof was used. Where and how could it be used? Justify the step-by-step process of the ZK protocol.
12. Figures 1 and 5 show the high-level logic of the model. Suggested to include or update them with a logical view.
13. Suggested to include a design diagram to show the flow of transactions among Ring LWE, Sharding, POS Validation, TSS, Atomic Swap Protocol, and ZK Proof.
14. Provide a clear explanation, concisely, about Blockchain features for what reason you have chosen to collaborate with the Post Quantum
15. How does this frame work enhance the protection of BCT network transaction data, which is already protected by a one-way hash by Merkle as well as Digital Signatures based on ECC?

Experimental design

1. The authors did not include the total dataset in the raw data. Are you considering only a 9-record dataset for your proposed framework? How could you justify that your system is efficient if not taken the big dataset?

2. Not shown the implementation logics of ZK Proof, Ring LWE, BCT Transaction data, etc

3. Simulation results were not showing the proof of the concepts addressed, ZK Proof, Ring LWE, and BCT Transaction data

Validity of the findings

.

---

## Round 0.2 · accepted · Accept

· Academic Editor

Accept

The paper may be accepted.

Reviewer 1 ·

Basic reporting

Accepatable

Experimental design

Acceptable

Validity of the findings

Acceptable

Additional comments

Authors revised paper carefully.

Reviewer 2 ·

Basic reporting

No objection, Considered all suggested comments

Experimental design

No objection, Considered all suggested comments

Validity of the findings

No objection, Considered all suggested comments

Additional comments

No objection, Considered all suggested comments